# Self-assembly of biomorphic carbon/sulfur microstructures in sulfidic environments

Julie Cosmidis[1] & Alexis S. Templeton[1]

In natural and laboratory-based environments experiencing sustained counter fluxes of sulfide and oxidants, elemental sulfur ($S^0$)—a key intermediate in the sulfur cycle—can commonly accumulate. $S^0$ is frequently invoked as a biomineralization product generated by enzymatic oxidation of hydrogen sulfide and polysulfides. Here we show the formation of $S^0$ encapsulated in nanometre to micrometre-scale tubular and spherical organic structures that self-assemble in sulfide gradient environments in the absence of any direct biological activity. The morphology and composition of these carbon/sulfur microstructures so closely resemble microbial cellular and extracellular structures that new caution must be applied to the interpretation of putative microbial biosignatures in the fossil record. These reactions between sulfide and organic matter have important implications for our understanding of $S^0$ mineralization processes and sulfur interactions with organic carbon in the environment. They furthermore provide a new pathway for the synthesis of carbon-sulfur nanocomposites for energy storage technologies.

---

[1] Department of Geological Sciences, University of Colorado, Boulder, Colorado 80309, USA. Correspondence and requests for materials should be addressed to J.C. (email: julie.cosmidis@colorado.edu).

Microorganisms can induce the formation of a great variety of minerals, either intracellularly or on cellular surfaces and extracellular structures such as sheaths, stalks, S-layers and pili[1,2]. The close association between these biominerals and their templating organic structures can serve as morphological and chemical biosignatures of specific microbial processes in the environment, and, if preserved during diagenetic and fossilization processes, throughout Earth's geological record[3–6].

Elemental sulfur ($S^0$) is an important intermediate in the biogeochemical cycle of sulfur[7], that can be formed as a biomineral by a wide diversity of photo- and chemotrophic microorganisms that can oxidize reduced sulfide species to $S^0$ and store $S^0$ intra- or extracellularly[8]. Elemental sulfur is considered a rather unstable and dynamic constituent of the sulfur pool of sediments[7,9], being only thermodynamically stable under a very restricted range of Eh and pH conditions[10]. In rare instances, $S^0$ does accumulate and form conspicuous deposits, generally at oxic/anoxic boundaries supplied by a flux of sulfide. The biogeochemical conditions conducive to the formation and preservation of $S^0$ in the environment are not fully understood. However, $S^0$ is frequently intimately associated with microbial cells, biofilms and extracellular organic materials, so that $S^0$ is often regarded as a biosignature of the activity of sulfide-oxidizing microorganisms[11–13]. $S^0$ is quite labile and is generally not preserved through diagenesis, but carbonaceous microstructures intimately associated with sulfides[14,15] or organically bound sulfur[16–18] have been described throughout the rock record and interpreted as fossils of sulfur-cycling bacteria.

Here we show that 'false biosignatures' can be produced abiogenically by reacting sulfide with dissolved organics. In this work, nanometer- to micrometre-sized carbonaceous filaments, tubes and spheres mineralized with $S^0$ were formed through a self-assembly process in sterile laboratory sulfide/oxygen gradient environments containing complex mixtures of organic compounds such as yeast extract and/or peptone. Our detailed characterization of these carbon/sulfur (C/S) microstructures shows that they possess morphologies and compositions notably similar to microbial cellular and extracellular structures associated with $S^0$, thus meeting the most robust criteria commonly used for identifying microbes and biominerals in the environment and in the fossil record. The formation of these C/S microstructures furthermore provides a new mechanism for $S^0$ formation and stabilization involving complex interactions with organic matter that might shed light on $S^0$ formation processes in the environment. This study also reveals new pathways for the synthesis of nano-materials with the potential for technological applications, such as the development of advanced nanostructured carbon–sulfur cathode materials for next-generation Li-S batteries[19–21]. The spectrum of recently developed synthesis methods for such composites are complex and require several energy-intensive steps, which is a barrier for large-scale fabrication and commercial utilization. Here we show that multidimensional carbon tubes and spheres loaded with $S^0$ can be produced in a simple one-step energy-efficient manner, which might improve considerably the technical and commercial feasibility of Li-S energy storage.

## Results

### Abiogenic formation of carbon/sulfur microstructures.
The C/S microstructures were produced in gradient tubes composed of an autoclaved agarose plug containing 5 mM of sodium sulfide, and an autoclaved aqueous top layer composed of a mineral medium amended with $0.125–10\,g\,l^{-1}$ yeast extract and/or peptone (Fig. 1a). Under these conditions, counter gradients of

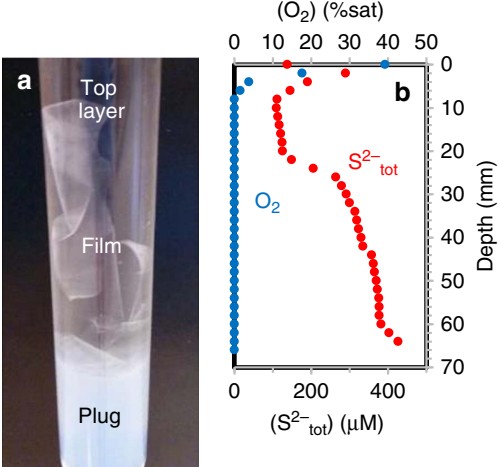

**Figure 1 | Sulfide gradient tubes for the synthesis of abiogenic carbon/sulfur microstructures.** (**a**) Photograph of a sulfide gradient tube showing the agar plug (white) and the aqueous top layer (clear) containing a whitish agarose film. (**b**) Typical total sulfide ($S^{2-}_{tot}$) and oxygen ($O_2$) profiles in the top layer of a tube after 1 day of experiment. Total sulfide concentrations were calculated using measured $H_2S$ and pH profiles (see Methods section).

oxygen and sulfide were rapidly established in the tubes (Fig. 1b; Supplementary Fig. 1). After 3–6 days, filamentous and spherical microstructures could be observed in the tubes where dissolved organics were present (Figs 2–5). These microstructures were either found suspended in the aqueous top layer, or associated with a whitish agarose film (Fig. 1a; Supplementary Fig. 2) that must be derived from the agar plug at the bottom of the tubes. No filamentous or spherical C/S microstructures were observed in control sulfide gradient tubes where dissolved organics were omitted.

### Diversity of carbon/sulfur microstructures.
A great diversity of C/S filamentous microstructures with a wide range of sizes, shapes and spatial organizations could be obtained depending on the concentration of yeast extract and/or peptone present in the top layer of the sulfide gradient tubes (Figs 2 and 3). The thicknesses of the filaments ranged from ∼40 nm to several micrometres, and the filaments could extend from a few tens of micrometres up to several hundreds of micrometres in length. The size of the structures depended on the type and concentration of dissolved organics present in the gradient tube, where the mean thickness generally increased with an increase in the concentration of yeast extract and/or peptone introduced in the tubes. Filament thicknesses also increased with time during the experiments. The filaments could be either flexible (Fig. 2b) or rigid (Figs 2c,d and 3c,d), branching (Figs 2d and 3c) or not branching (Fig. 3d) and were isolated, organized in bundles (Fig. 2b) or connected in perpendicular arrays (Fig. 2c). Branching frequently occurred at 45° and 90° angles (Figs 2c,d and 3c). Helical filaments were sometimes present in mixture with rectilinear filaments (Fig. 3f). The spherical C/S microstructures were always observed in association with the filaments (as for instance on Fig. 3d). Their diameters did not depend on the composition or concentration of organics and always ranged from 200 to 2,000 nm. They were delimited by a fragile shell that sometimes was broken, showing that some of those spheres were empty (Fig. 3e). As mentioned previously, no spherical or filamentous C/S microstructures were found in sulfide gradient

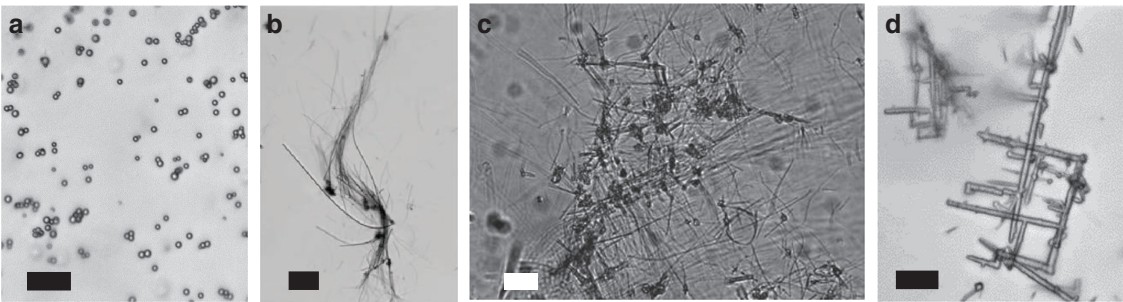

**Figure 2 | Light microscopy images of structures formed in sulfide gradient tubes under variable concentrations of dissolved organics.** (**a**) Rounded mineral sulfur grains formed in the absence of organics. (**b**) Bundling flexible filaments formed in the presence of yeast extract $2\,\mathrm{g\,l^{-1}}$. (**c**) Rigid filaments forming a perpendicular two-dimensional lattice, formed in the presence of yeast extract $5\,\mathrm{g\,l^{-1}}$ and peptone $5\,\mathrm{g\,l^{-1}}$. (**d**) Rigid branching filaments formed in the presence of yeast extract $2\,\mathrm{g\,l^{-1}}$ and peptone $10\,\mathrm{g\,l^{-1}}$. Scale bars, $10\,\mathrm{\mu m}$ (**a**); $20\,\mathrm{\mu m}$ (**b–d**).

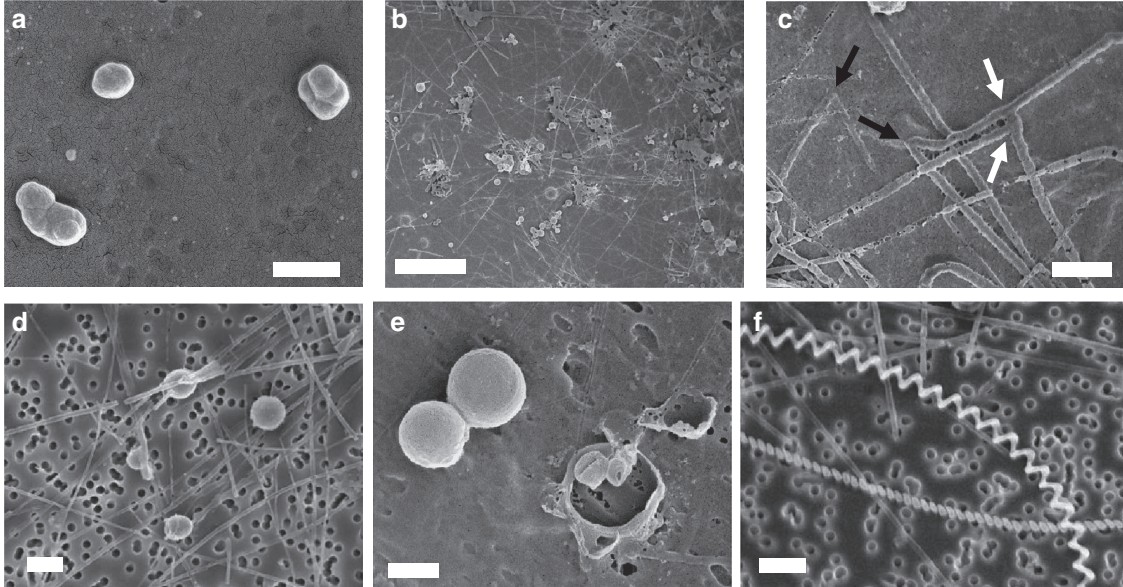

**Figure 3 | SEM images in the SE mode of structures formed in sulfide gradient tubes under variable concentrations of dissolved organics.** (**a**) Rounded mineral sulfur grains with irregular shapes formed in the absence of organics. (**b**) Large view of C/S microstructures formed in the presence of yeast extract $0.125\,\mathrm{g\,l^{-1}}$ and peptone $0.625\,\mathrm{g\,l^{-1}}$ showing that the filaments form a two-dimensional network. (**c**) Close-up on filaments branching at 45° (black arrows) and 90° (white arrows) angles. (**d**) Rectilinear filaments and spheres formed in the presence of yeast extract $2\,\mathrm{g\,l^{-1}}$. (**e**) Spheres formed in the presence of yeast extract $0.125\,\mathrm{g\,l^{-1}}$ and peptone $0.625\,\mathrm{g\,l^{-1}}$. The fragile shell of the spheres on the right part of the image is broken, showing that these spheres are empty. (**f**) Helical filaments and spheres formed in the presence of yeast extract $2\,\mathrm{g\,l^{-1}}$. Scale bars, $1\,\mathrm{\mu m}$ (**a,c,d–f**); $10\,\mathrm{\mu m}$ (**b**).

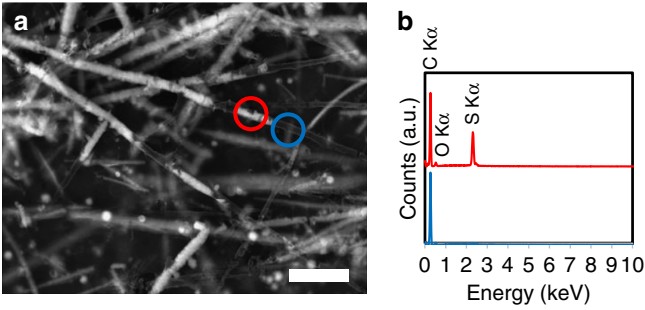

**Figure 4 | SEM image in the BSE mode and EDXS analyses of carbon/ sulfur microstructures.** (**a**) BSE SEM image of filaments and spheres formed in the presence of yeast extract $2\,\mathrm{g\,l^{-1}}$ and peptone $10\,\mathrm{g\,l^{-1}}$, showing that the almost transparent carbon shells are partially filled with sulfur (bright phase). (**b**) EDXS spectra of the areas circled in **a** showing C and C/S regions. Scale bar, $10\,\mathrm{\mu m}$ (**a**).

tubes prepared without dissolved organics. Under these conditions, small amounts of $S^0$ were instead formed as rounded mineral grains with irregular shapes (Figs 2a and 3a; Supplementary Fig. 3b). Unlike spherical C/S microstructures, these mineral grains were not surrounded by a shell.

**Ultrastructure of the carbon/sulfur filaments and spheres.** Scanning electron microscopy (SEM) imaging in the back-scattered electron (BSE) mode coupled with energy-dispersive X-ray spectroscopy (EDXS) provided further insight into the ultrastructure and composition of the C/S microstructures. The filamentous and spherical C/S microstructures are formed by an almost electron-transparent shell mostly composed of carbon (with minor amounts of oxygen). The C shells of the micro-structures are then partially filled by a bright phase composed of sulfur (Fig. 4). This ultrastructure was resolved at higher resolu-tion by transmission electron microscopy (TEM) imaging, also

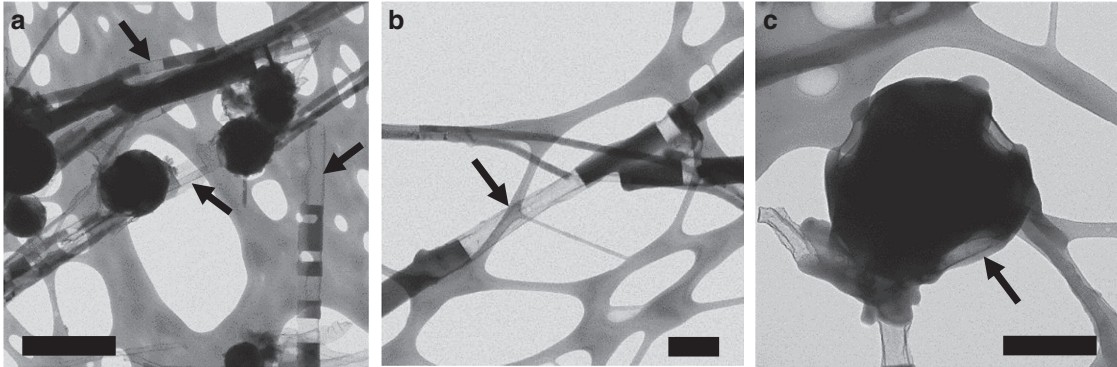

**Figure 5 | TEM images of carbon/sulfur microstructures on a lacey carbon TEM grid.** The C/S microstructures were formed in the presence of yeast extract $2 \, g \, l^{-1}$. Sulfur (the dark electron-dense phase) partially fills the more transparent (lighter grey) tubular and spherical carbon shells (black arrows). Scale bars, 500 nm (**a**); 200 nm (**b,c**).

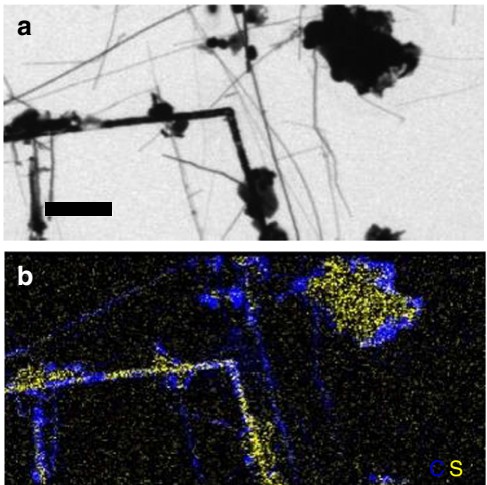

**Figure 6 | STXM mapping of the carbon/sulfur microstructures at the C K-edge and the S L-edge.** (**a**) Image at 288.2 eV of C/S microstructures obtained in the presence of yeast extract $2 \, g \, l^{-1}$. (**b**) Map of carbon (blue) and sulfur (yellow) obtained on the same area as **a**. Scale bar, 5 μm (**a**).

showing that sulfur, the dark electron-dense phase on Fig. 5 images, partially fills the inside of the tubular or spherical carbon shells of the microstructures.

**Chemical speciation of sulfur and carbon.** Scanning transmission X-ray microscopy (STXM) at the C K-edge and S L-edge was used to characterize the distribution and speciation of carbon and sulfur in the C/S microstructures at the submicron scale. STXM maps of C and S confirmed the close association between these two elements in the microstructures (Fig. 6).

The sulfur phase could be identified as $S^0$ in the cyclooctasulfur form ($S_8$) based on bulk X-ray absorption near-edge structure (XANES) spectroscopy at the S K-edge (Fig. 7), as well as Raman spectromicroscopy (Supplementary Fig. 3) and STXM/XANES analyses at the S L-edge (Supplementary Fig. 4).

S K-edge XANES spectra presented in Fig. 7 allow a qualitative comparison of the amount of $S^0$ produced in the absence and the presence of organic matter. The S K-edge XANES spectra of all the samples analysed present peaks at 2,472.6 and 2,479.3 eV, corresponding to $S^0$, as well as peaks at 2,481.2 and 2,482.6 eV, corresponding to sulfonate and sulfate groups that are present in the polycarbonate filters on which the samples were deposited for the analyses. The relative contributions of these two components

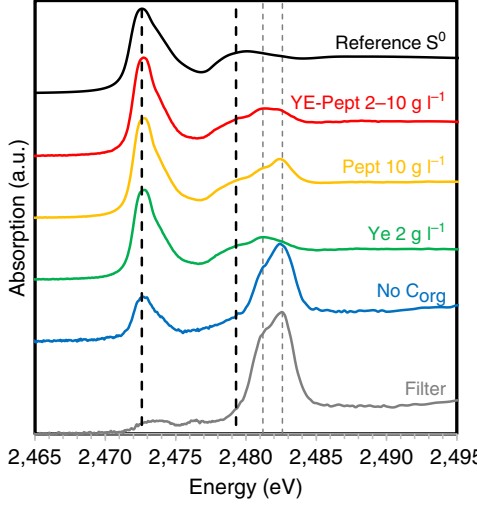

**Figure 7 | Bulk S K-edge XANES analyses of carbon/sulfur microstructures obtained in the presence of different concentrations of yeast extract and/or peptone.** The spectra of a reference $S^0$ as well as the polycarbonate filters on which the samples were deposited are also shown (see Methods). Black vertical dashed lines correspond to 2,472.6 and 2,479.3 eV (main peaks in the reference $S^0$ spectrum), whereas grey vertical dashed lines correspond to 2,481.2 and 2,482.6 eV (sulfonate and sulfate groups present in the polycarbonate filters).

($S^0$ versus filter) gives a comparative estimate of the amount of $S^0$ present in the samples. It is apparent that a smaller quantity of $S^0$ was formed in the gradient tube without any organics compared with the gradient tubes containing yeast extract and/or peptone.

The carbon shells of the microstructures present complex organic compositions, as determined by STXM/XANES analyses at the C K-edge (Fig. 8). Their spectra present peaks at 285.0 and 285.2 eV ($1 \, s \to \pi^{\star}_{C=C}$ transitions of unsaturated or aromatic C), 287.3 eV ($1 \, s \to \sigma^{\star}$ transitions of aliphatic C) and 288.2 eV ($1 \, s \to \pi^{\star}_{C=O}$ transitions in amide groups)[6,22–24], and are relatively similar to the spectra of the yeast extract and peptone from which they were derived (Fig. 8e). However, the C K-edge spectra of the microstructures do differ from the spectrum of the agarose film with which they are associated. More interestingly, the complex C K-edge composition of the organic carbon shell of the C/S microstructures is very similar to the to the composition of microbial cells, represented here by the spectrum of the bacterium *Escherichia coli*[25], as well as extracellular twisted stalks produced by the Fe-oxidizing bacterium *Mariprofundus ferrooxydans*[24].

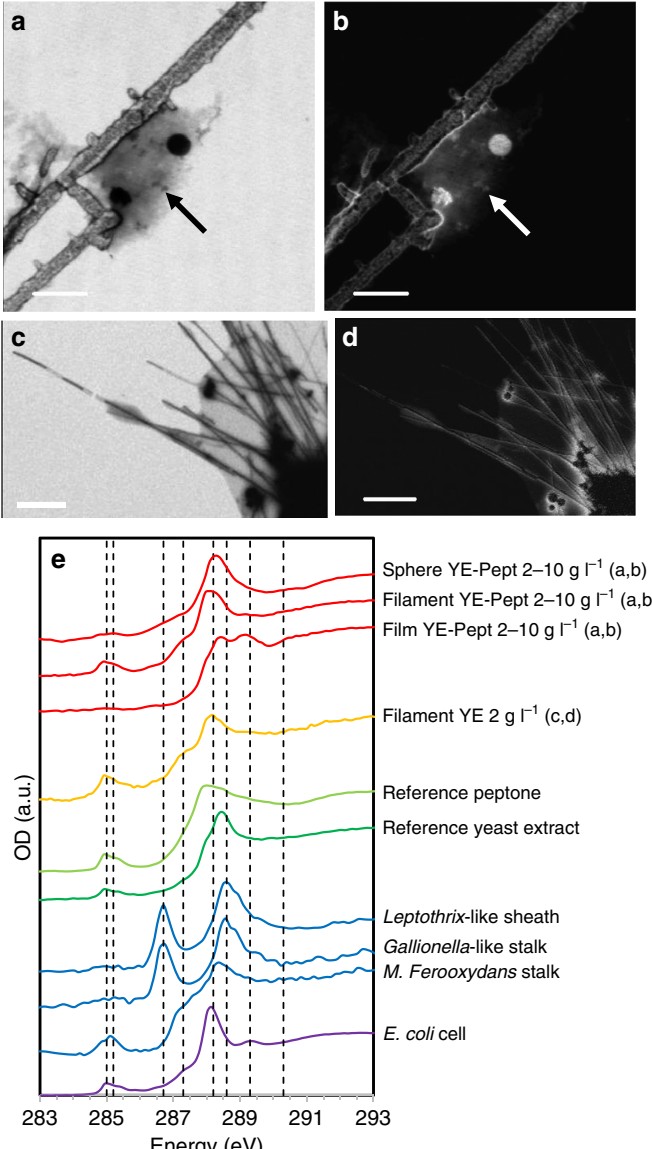

**Figure 8 | C K-edge STXM/XANES analyses of carbon/sulfur microstructures.** (**a**,**c**) Images at 288.2 eV and (**b**,**d**) organic C maps of C/S microstructures obtained in the presence of yeast extract 2 g l$^{-1}$ and peptone 10 g l$^{-1}$ (**a**,**b**), and yeast extract 2 g l$^{-1}$ (**c**,**d**). Arrows in **a** and **b** point towards a fragment of the agarose film attached to the microstructures. (**e**) C K-edge STXM/XANES spectra of C/S microstructures shown in **a**–**d** as well as reference spectra for yeast extract, peptone, microbial cells and different microbial extra-cellular structures. Dashed lines indicate energies at 285.0 (1 s→π*$_{C=C}$ transitions of unsaturated or aromatic C), 285.2 (1 s→π*$_{C=C}$ transitions of unsaturated or aromatic C), 286.8 (1 s→π*$_{C=O}$ transitions in ketone or aldehyde functional groups), 287.3 (1 s→σ* transitions of aliphatic C), 288.2 (1 s→π*$_{C=O}$ transitions in amide groups), 288.6 (1 s→π*$_{C=O}$ transitions in carboxylic groups), 289.3 (1 s→3p/σ* transitions in alcohols, ethers or hydroxylated aliphatics) and 290.3 eV (1 s→π*$_{C=O}$ transitions in carbonate groups)[6,22–24]. The spectra of the C/S microstructures are similar to those of yeast extract, peptone, *Escherichia coli* cells[25] and *Mariprofundus ferrooxydans* twisted stalks[24], but differ from the spectra of sheaths and twisted stalks of other environmental Fe-oxidizing bacteria[23]. Scale bars, 5 μm (**a**–**d**). OD, optical density.

## Discussion

The formation of carbon tubes and spheres loaded with S$^0$ through the interaction of dissolved organic carbon, oxygen and

sulfide has not been previously observed or expected. It is particularly surprising that the growth of the particulate C/S microstructures is purely abiogenic and does not require any microbial activity. The organic compounds used for the formation of the C/S microstructures were biologically derived, so that the use of the term 'abiogenic' could be debated. Indeed, yeast extract and peptone are complex mixtures of amino acids, peptides, carbohydrates and vitamins obtained from the enzymatic hydrolysis of yeast cells and animal tissue, respectively. Further experiments using simpler non-biological (for example, prebiotic) organic compounds are currently being performed and their results will be described in a future study. The term 'abiogenic' is used here to refer to experimental conditions where no microbial or enzymatic activity is invoked. Indeed, a diversity of C/S composite materials could be repeatedly produced in aseptic (autoclaved) gradient tubes that were not inoculated and that did not show evidence of microbial cells (see Methods).

The formation of C/S microstructures clearly involves the self-assembly of dissolved organic molecules into carbonaceous scaffolds and membranes that encapsulate S$^0$. C/S microstructures were not observed in experiments where no sulfide was added to the agar plug, demonstrating that sulfide is required for the self-assembly process, potentially by cross-linking of the organic molecules. The dissolved organics also play an important role in the oxidation of sulfide and/or stabilization of S$^0$ as an oxidation product of sulfide oxidation. Hydrogen sulfide was indeed consumed faster in the experiments where organics were present (Supplementary Fig. 1), and larger quantities of S$^0$ were produced in the presence of organics compared with organic-free experiments (Fig. 7). This could be partly due to the presence of oxidized functional groups in the dissolved organic matter, which have been shown to serve as electron acceptors for the oxidation of hydrogen sulfide at rates competitive to sulfide oxidation by molecular oxygen or iron oxides[26,27]. A catalytic influence of organics on sulfide oxidation has also been described previously in a study showing that reaction of aldehydic carbonyl groups with FeS can control the chemistry, mineralogy and quantity of oxidized iron sulfides that are produced[28].

The existing knowledge regarding the reaction of reduced sulfur species with organic matter in the environment is almost exclusively focused on organic matter sulfurization, that is, the formation of organosulfur compounds on incorporation of sulfur into organics[29]. The presence of organosulfur compounds was not detected in our experiments by bulk S K-edge XANES or S L-edge STXM/XANES analyses, either because they are not present in appreciable quantities, or because the total sulfur pool is dominated by the S$^0$ co-precipitated in the C/S microstructures. In contrast, our study shows that complex mineralization reactions between organics and sulfide can be quantitatively significant.

The experimental conditions used here (simple mineral medium, circumneutral pH, ambient temperature, low oxygen concentration and presence of organics and sulfide) are relevant to natural geochemical environments where S$^0$ commonly forms, which present oxic/anoxic interfaces exposed to sustained fluxes of sulfide and are commonly rich in particulate and dissolved organic carbon from diverse sources[7]. Typical total sulfide concentrations in the aqueous overlayer in these gradient tube experiments are 25–400 μM (Supplementary Fig. 1), which is comparable to sulfide concentrations in sulfidic environments where S$^0$ can be found, such as marine sediments[7,30], sulfidic caves[31], euxinic lakes[32] or sub-glacial springs[11]. Dissolved organic carbon concentrations used in our experiments are one to four orders of magnitude higher than those commonly found in environmental systems. Future experiments will be conducted using lower organic concentrations to determine the rate of C/S

formation and the morphological diversity of the C/S composites produced under such conditions.

Altogether, we suggest that the formation of C/S materials may be a ubiquitous process at oxic/anoxic boundaries that experience sulfide fluxes in the presence of dissolved organic matter, and provides a previously overlooked process for the environmental formation of $S^0$. Sulfide oxidation and $S^0$ stabilization by intracellular or excreted microbial organic compounds could for instance provide a quantitatively important pathway to immobilize sulfur in the environment, which can then be retained (as solid-phase $S^0$) as an energy source for microorganisms rather than lost (as dissolved hydrogen sulfide) by diffusion or advection. Consideration of such processes might for instance explain the presence of organic compounds associated with sulfur in the intracellular sulfur globules produced by some bacteria[33], as well as shed new light on the mechanism of microbial filamentous sulfur formation at hydrothermal vents[34,35].

Natural environments rich in $S^0$ often host complex communities of sulfur-cycling microorganisms, including filamentous bacteria, and therefore abiogenic C/S filamentous or spherical microstructures could often be misidentified as bacteria or extracellular microbial structures such as biomineralized sheaths. C/S microstructures might also have been formed in other experimental systems where organics and sulfide where allowed to react, but the reaction products might either not have been characterized microscopically, or have been misinterpreted as microbial. In this respect, it is intriguing to revisit previous sulfide gradient tube cultivation experiments conducted by Gleeson et al.[36] In these experiments, sulfide gradient tubes similar to those used in the present study were used and inoculated with microbial consortia enriched from sulfur deposits at Borup Fiord Pass (Canadian High Arctic). In these experiments, Gleeson et al.[36] observed filamentous microstructures mineralized with $S^0$, that they called 'filaments' (a few hundred nm across) and 'sheaths' (measuring ∼1 μm across). These filamentous microstructures were frequently attached at one end of their length and radiating out from a central mass, as sometimes observed in our experiments (compare C/S microstructures in Fig. 2b and Supplementary Fig. 3c with 'filaments' and 'sheaths' in ref. 36). These 'filaments' and 'sheaths' were not produced in Gleeson et al.'s non-inoculated gradient tubes, which led them to interpret them as biomineralized microbial structures. In light of our work here, we propose that these authors observed abiogenic C/S microstructures similar to those described in the present study. Although Gleeson et al. did not supplement their gradient tubes with organics such as yeast extract and peptone, we can propose that C/S microstructure formation in this case resulted from the interaction of sulfide with low concentrations of dissolved organics derived from the tube's inoculum (for example, excreted by the agar-digesting bacterial consortia). It was not until the work described herein that we determined that such structures could be produced without inoculating such gradient tubes, so long as such abiotic experiments were amended with dissolved organic carbon.

C/S microstructures similar to those formed in our experiments might also have formed in past environments where both dissolved organics and sulfide were present. Such microstructures could have been preserved in the rock record and potentially misinterpreted as microbial biosignatures, due to their morphological and compositional resemblance with microbial cellular and extracellular structures. The C/S filamentous and spherical microstructures are indeed strikingly biomorphic, leading us to reconsider existing definitions of biogenicity criteria as proposed by Brasier and Wacey[4]. The first of these criteria is 'a cell-like morphospace'. The C/S microstructures present morphologies and dimensions similar to microbial cells such as filaments and

cocci, and extracellular structures such as outer membrane vesicles formed in microbial biofilms[37,38], as well as microbial sheaths and twisted stalks produced by Fe-oxidizing bacteria[23]. The twisted morphology of the latter in particular, combined with their organic composition, are considered a robust biosignature of microbial Fe-oxidation in the geological record[6,24]. The second biogenicity criterion defined by Brasier and Wacey[4] is 'a biology-like behaviour'. The spatial organization of the spherical C/S microstructures in clusters (Fig. 3e) and of the filamentous C/S microstructures in complex two-dimensional networks (Fig. 3b) mimics the tendency for microbial cells to form spheroid clusters and mats of intertwined filaments. The third criterion is 'a metabolism-like behaviour'. The C/S microstructures are delimited by a membrane-like carbon shell with a complex organic composition (Fig. 8). Their C K-edge STXM/XANES spectra is in particular dominated by the $1 \, s \rightarrow \pi^\star$ transitions in amide groups, which is also the main feature the C K-edge STXM/XANES spectra of microorganisms[22,23,39], and has thus often been proposed as a biogenicity criterion for the identification of microbial remains in the fossil record[5,40–42]. The C/S microstructures moreover contain $S^0$, a mineral commonly formed as a by-product of the metabolism of sulfide-oxidizing bacteria and that can be stored intracellularly by the cells[8]. Finally, the last biogenicity criterion defined by Brasier and Wacey[4] is 'a viable context for life'. As discussed previously, the C/S microstructures are formed under environmentally relevant experimental conditions that are furthermore suitable for life (room temperature, circumneutral pH, presence of carbon sources and nutrients, disequilibrium between reductants and oxidants). Therefore, the abiogenic formation of C/S microstructures shown in this study challenges commonly accepted biogenicity criteria, whether based on morphology, spatial organization or chemical composition.

In assessing the dimensions of the abiogenic C/S microstructures, we note that in our experiments containing the lowest dissolved organics concentrations, which are the closest to environmental dissolved organic carbon abundances, C/S filaments are often significantly thinner (a few tens of nanometres across) than filamentous bacteria or microbial sheaths (a few hundreds of nanometres to several micrometres across). However, our experiments were conducted using yeast extract and peptone as sources of dissolved organic carbon, but one could expect that with other types of organic compounds, the organic concentration-filament size relationship might be different and must be further explored.

Previous work by García-Ruiz et al.[43–45] showed the abiogenic formation of micron-sized silica-carbonate filaments, called biomorphs, exhibiting complex morphologies similar to Precambrian putative microfossils. The twisted C/S microstructures shown in Fig. 3f are in particular reminiscent of García-Ruiz et al.'s helical biomorphs (for example, Fig. 2 in ref. 44). However, the results from this study call for an even greater re-evaluation of biogenicity criteria, since the carbon–sulfur 'biomorphs' produced here are organic-mineral microstructures, rather than purely mineral (for example, silica-carbonate biomorphs), thus mimicking ever more closely microbial remnants. Our discovery of such 'false biosignatures' calls even stronger caution in interpretation of putative microfossils in the rock record.

It is particularly notable that the C/S microstructures formed in our experiments are very similar to putative microfossils found in a diversity of organic and sulfide-rich rocks from different ages and locations, including the hollow spheroidal and tubular microstructures delimited by carbonaceous cell walls described in sandstones of the ∼3.4 Ga Strelley Pool Formation in Western

Australia[14]. They are particularly reminiscent of the recently described carbonaceous filaments present in ~1.8 Ga cherts of the Duck Creek Formation[15], which were interpreted as fossils of sulfur-cycling microorganisms. We propose that, rather than establishing direct evidence for biogenicity or the presence of sulfur-cycling microbial communities, carbonaceous micro-structures associated with sulfur might result from abiogenic reactions in sulfidic systems containing dissolved organic compounds. If criteria can be identified that allow us to rigorously differentiate abiogenic C/S microstructures from microbial fossils, then these microstructures could alternatively serve as critical indicators of past environmental conditions, including the detection of environments that were far from equilibrium. Such information is highly valuable for recon-structing past redox interfaces on Earth, as well as in seeking habitable environments on other planetary bodies that may undergo active sulfur cycling, such as Jupiter's moon Europa. Thus we caution that 'sulfur microfossils' similar to the C/S microstructures should not be used as targets for life-detection, but they are still valuable for reconstructing local environmental processes.

The formation mechanism of the C/S microstructures should now be interrogated at the molecular level. In particular, future studies will have to determine by which process dissolved organic molecules self-assemble into the tubular and spherical organic membrane shells of the microstructures. The C K-edge STXM/XANES analyses of the C/S microstructures suggest a complex organic composition dominated by peptidic molecules (Fig. 8e). A great diversity of amphiphilic organic molecules, including peptidic amphiphiles, are known to self-assemble in aqueous media to form spheres as well as tubes via helical intermedi-ates[46,47]. Helical morphology in particular, which is sometimes observed in our experiments (Fig. 3f), might result from the close packing of chiral amphiphiles during self-assembly[47]. Further analyses will be required to determine the nature of the molecules involved here.

Finally, this study demonstrates a new pathway for the production of nanostructured sulfur–carbon composites inten-sively sought for energy storage applications. Rechargeable Li–S batteries, using sulfur at the cathode, are considered a promising successor to lithium-ion batteries that are powering most of today's portable electronic devices. Li-S batteries possess a theoretical energy density (2,600 Wh kg$^{-1}$) as much as five times higher than commercial batteries, and a theoretical capacity (1,675 mAh g$^{-1}$) an order of magnitude higher[48]. However, Li-S batteries present several technical issues that hinder their practical realization, such as the low electrical conductivity of the sulfur composing the cathode of the batteries, the loss of material from the cathode over many charge–discharge cycles by formation of highly soluble Li-polysulfides intermediates, and volume expansion and morphological disruption of the sulfur cathode by precipitation of insoluble products such as $Li_2S_2$ and $Li_2S$. Much of the recent research and development effort in the last decade has been focused on overcoming these drawbacks by developing new cathodes where sulfur is embedded within conductive carbon materials, which helps enhance the electrical conductivity of the cathode, limits the loss of intermediate sulfur species in the electrolyte solution, and accommodates volume changes[19,49]. The geometry of the C/S microstructures produced in this study, where sulfur is totally encapsulated in the carbonaceous tubes and spheres, is predicted to enable good confining of the polysulfides and to offer a large surface area for their absorption, potentially slowing down material loss from the cathode. In addition, the negative effect of volume changes on cathode stability over many charge–discharge cycles should be softened for our C/S microstructures, since the sulfur in these composites only partially fill the carbon tubes and spheres, leaving free space for volume expansion. Finally, a high sulfur content of the cathode is important to achieve high energy density and meet the future commercial practice requirements of Li–S batteries. It can be estimated from SEM imaging that some of our C/S microstructures possess a sulfur content ranging from 75 to 95 wt%. These values are very good compared with most existing sulfur–carbon composites, for which the sulfur loading ranges from 30 to 75 wt% and rarely reaches 80 wt% (ref. 19).

Existing methods for the fabrication of carbon–sulfur compo-sites for Li–S batteries are usually complex and typically involve several steps: first, the synthesis of the carbon material (for example, graphene sheets, carbon nanotubes, nanospheres, porous carbon, organic polymers and so on[20,21]), in some cases through high temperature (450–1,250 °C) methods such as arc discharge, vapourization using a laser, pyrolysis or chemical vapour deposition of hydrocarbons, and second, sulfur loading of the carbon material, often by sulfur melt adsorption or vapour phase infusion, also performed at high temperature. Compared with existing synthesis methods, the advantages of the method described herein are its practical simplicity, low cost and energy efficiency, particularly since the reactions are performed at room temperature. Moreover, the present method enables a high tunability of the morphology, size, sulfur content and three-dimensional organization of the C/S microstructures that can be obtained, which is also particularly attractive for the proposed application as cathodes of Li–S batteries, as it provides a control on the critical electrochemical properties and cycling performance of these materials.

We showed that the efficient synthesis of C/S composites can be achieved by mimicking natural systems with strong sulfide gradients. Once the electrical properties of such materials are well characterized, it will be critical to evaluate the roles such C/S composites may play in energy transfer in environmental systems as well.

## Methods

**Gradient tube experiments.** The method used to prepare the sulfide gradient tubes was modified from ref. 36. We used an artificial mineral medium (EM medium) containing only 10% of the NaCl in the original EM recipe (that is, 2.75 g l$^{-1}$ instead of 27.5 g l$^{-1}$), and with a pH adjusted to 7.5. The tubes consisted of an agar plug (modified EM medium + 1% w/v agar) containing 5 mM Na$_2$S, and an overlying column of modified EM medium supplemented with dissolved yeast extract (Fischer Scientific, CAS Number: 8013-01-2) and/or peptone (Fisher Scientific, CAS Number 73049-73-7) at concentrations ranging from 0.125 to 10 g l$^{-1}$, and sparged with N$_2$:CO$_2$ (80:20) for 20 min to de-oxygenate it. Both the top and bottom layer solutions and glass tubes were autoclaved prior to preparing the gradient experiments, and all the preparation was performed under aseptic conditions. The gradient tubes were stored overnight at 4 °C and then left in the dark at room temperature without inoculation. Samples were collected at different time points for analyses, always under aseptic conditions. The absence of microorganisms in the sulfide gradient tubes was confirmed by fluorescence microscopy using syto-9 and propidium iodide as fluorescent dyes that will stain viable and dead cells respectively (LIVE/DEAD Bacterial Viability Kit, Molecular Probes). The sterility of the gradient tubes was also checked by collecting aliquots from the top layer and spreading those on LB agar plates. No colonies were observed after several months of incubation at room temperature, so that contamination of the tubes by microorganisms can be ruled out. In addition, multiple generations of gradient tube experiments have been conducted using new solutions and reagents, showing that the abiogenic formation of the C/S microstructures can be reliably reproduced.

**Geochemical gradients measurements.** Vertical oxygen and hydrogen sulfide concentration profiles along the top layer of the gradient tubes were measured using Unisense (Aarhus, Denmark) OX-100 and H2S-100 microsensors (respectively). The oxygen microsensor was calibrated using freshly prepared de-oxygenated water (0% oxygen saturation) as well as a well-aerated water (100% oxygen saturation). The hydrogen sulfide solution was calibrated using freshly prepared de-oxygenated solutions of Na$_2$S in an acetate buffer (pH <4), in the 0–500 μM H$_2$S range. The microsensors as well as a calibrated pH micro-electrode (Thermo Scientific Orion PerpHecTROSS Combination pH Micro Electrode) were mounted on a motor to measure vertical H$_2$S, O$_2$ and pH profiles

in the top layer of the gradient tubes with a 2 mm step. We checked that profile measurements did not significantly perturb the chemical gradients in the tubes by measuring the same tubes after a few minutes interval and obtaining identical profiles. $H_2S$ and pH measurements were used to calculate total sulfide concentrations in the tubes using the acid dissociation constant for $H_2S/HS^-$ from ref. 50.

**Light and fluorescence microscopy.** Light and fluorescence microscopy was carried out with a Zeiss Axio Imager Z1 on samples from the sulfide gradient tubes. The samples were stained using syto-9 and propidium iodide dyes to check for the presence of bacteria.

**Scanning electron microscopy.** Samples from the gradient tubes were rinsed three times in distilled water, and deposited on a polycarbonate filter (GTTP Isopore membrane filters, Merck Millipore, pore size 0.2 μm). The filters were allowed to dry at ambient temperature and were coated with carbon or gold prior to SEM. The analyses were conducted on a JSM-7401F field emission SEM at the Nanoscale Fabrication Laboratory at the University of Colorado at Boulder. Images were acquired in the secondary electron (SE) mode with the microscope operating at 3 kV and a working distance (WD) of ~3 mm, and in the BSE mode, at 10 kV and WD ~8 mm. EDXS analyses were performed at 20 kV and WD ~8 mm.

**Transmission electron microscopy.** Samples for TEM were rinsed three times in distilled water and concentrated by centrifugation. The samples were deposited on a lacey Formvar/Carbon TEM grid and allowed to settle for a few minutes before the grids were gently blotted with filter paper. Images were acquired using a FEI Tecnai Spirit BioTwin LaB₆ TEM operating at 100 kV equipped with an AMT XR41 digital camera.

**Fourier transform infrared spectromicroscopy.** Samples collected from the gradient tubes were deposited on a $CaF_2$ plate and rinsed three times with deionized water. Fourier transform infrared spectromicroscopy (FTIR) analyses were performed using a Thermo Nicolet Continuum microscope linked to a Nexus 670 FTIR spectrometer. The infrared beam was collimated by a $100 \times 100$ μm window and focused on the samples. Infrared absorbance spectra were collected on a liquid nitrogen-cooled MCTA detector to minimize electronic noise and water absorption in the detector. Spectra were acquired in transmitted mode between 4,000 and $400\,cm^{-1}$, and each spectrum obtained represents an integration of 150 spectral scans, with a wavenumber resolution of $4\,cm^{-1}$. Background corrections were applied to the data following each measurement to compensate for instrumental noise and contributions from atmospheric $CO_2$ and $H_2O$ by dividing the absorbance of the sample spectrum by the background spectrum at each data point. The spectrum of an agarose reference powder (Fisher Scientific) was acquired using the same method.

**Raman spectromicroscopy.** Raman spectromicroscopy was performed on samples from the gradient tubes deposited on a $CaF_2$ plate using a Horiba LabRAM HR Evolution Raman spectrometer equipped with a 532 nm frequency-doubled Nd:YAG laser (Laser Quantum) coupled to an Olympus BXFM optical microscope. The laser beam was focused through a 50x objective lens, yielding a spatial resolution of ~2 μm. A 600 lines per mm grating and adjustable confocal pinhole (100–200 μm) was used to give a spectral resolution full width at half maximum (FWHM) of $4.5–8.4\,cm^{-1}$. Spectra were collected and processed using LabSpec 6 software (Horiba Scientific) and a Si-based CCD detector ($1,024 \times 256$ pixels). The spectrometer was calibrated using the $520\,cm^{-1}$ Raman peak of Si prior to analysis. Spectral data were corrected for instrumental artifacts and baseline-subtracted using a polynomial fitting algorithm in LabSpec 6. The samples spectra were compared with a reference Raman spectrum for elemental sulfur (RRUFF database[51]).

**XANES spectroscopy.** Samples for bulk XANES spectroscopy were collected after 3 days of experiment. For each sample, 10 ml of experimental medium was passed through a polycarbonate filter (GTTP Isopore membrane filters, Merck Millipore, pore size 0.2 μm). The materials collected on the filters were then rinsed three times in distilled water. The filters were then immediately placed at $-20\,°C$ and kept frozen until analyses. XANES spectra were collected at the S K-edge on beamline 4-3 at Stanford Synchrotron Radiation Lightsource. A liquid He cryostat was used to keep the samples frozen during XANES analyses. Between 2 and 3 spectra were accumulated for each sample to improve the signal-to-noise ratio. Energy calibration was performed at several intervals between samples using the S K-edge spectrum of a sodium thiosulfate standard, setting the position of the maximum of the first pre-edge feature to an energy of 2,472 eV (ref. 52). Two to four spectra were acquired and averaged per sample. A linear background determined in the pre-edge region (2,400–2,470 eV) was subtracted from the averaged data, and the spectra were then normalized at 2,510 eV. S K-edge XANES spectra of a polycarbonate filter and of a standard $S^0$ compound (precipitated sulfur, Fisher

Scientific) on sulfur-free polypropylene tape were also acquired and similarly processed.

**Scanning transmission X-ray microscopy.** Samples harvested at different time of the experiments were rinsed three times in deionized water and a small drop (~1 μM) was deposited on a Formvar-coated TEM grid and allowed to dry at ambient temperature. STXM analyses were performed on beamline 10ID-1 (SM) of the Canadian Light Source (Saskatoon, Canada). The X-ray beam was focused on the samples using a Fresnel zone plate objective and an order-sorting aperture yielding a focused X-ray beam spot of ~30 nm. After sample insertion in STXM, the chamber was evacuated to 100 mtorr and back-filled with He at ~1 atm pressure. Energy calibration was achieved using the well-resolved 3p Rydberg peak of gaseous $CO_2$ at 294.96 eV. Images, maps and image stacks were acquired in the 260–340 eV (C K-edge) and 155–190 eV (S L-edge) energy ranges. The aXis2000 software (unicorn.mcmaster.ca/aXis2000.html) was used for data processing. A linear background correction was applied to the spectra at the C K and S L-edge, in the 260–280 eV region and 155–160 eV region, respectively, to eliminate the contribution of lower energy absorption edges.

Maps of organic C were obtained by subtracting from an image at 288.2 eV ($1\,s \to \pi^*$ electronic transitions in peptides[22]) and converted into optical density (OD) an image obtained at 280 eV (that is, below the C K-edge) and converted into OD. Maps of S were obtained by subtracting from an image at 163.5 eV (energy of the S $L_3$-edge) and converted into OD an image obtained at 160 eV (that is, below the S L-edge) and converted into OD. STXM-based XANES (STXM/XANES) spectra were extracted from image stacks as explained in ref. 39. The S L-edge spectrum of a reference $S^0$ compound (precipitated sulfur, Fisher Scientific) as well as the C K-edge spectra of reference particulate yeast extract (Fischer Scientific, CAS Number: 8013-01-2) and peptone (Fisher Scientific, CAS Number 73049-73-7) were also obtained using the same method. The C K-edge spectra of the samples were compared with spectra obtained on *E. coli* cells[25], *Mariprofundus ferooxydans* stalks[24], as well as environmental twisted stalks reminiscent of *Gallionella ferruginea* (*Gallionella*-type stalks) and sheaths reminiscent of *Leptothrix ochracea* (*Leptothrix*-type sheaths)[23].

**Data availability.** The authors declare that the data supporting the findings of this study are available within the article and its Supplementary Information Files.

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

## Acknowledgements
This work was supported by NASA Exobiology grant NNX13AJ32G and the NASA Astrobiology Cooperative Agreement NNA15BB021. We also thank the Provost's office at the University of Colorado for support, and the University of Colorado Technology Transfer Office for their assistance in developing a patent application that encompasses the work described herein. We acknowledge Graham Lau (University of Colorado) and Dr Katherine Wright (Natural Environment Research Council, UK) for their collaboration prior to this study, Mary Morphew for her help with TEM analyses (Boulder Electron Microscopy Services, University of Colorado), and Eric Ellison for his technical support during Raman spectroscopy analyses (Raman Microspectroscopy Laboratory at the Department of Geological Sciences, University of Colorado). SEM analyses were conducted at the Nanomaterials Characterization Facility, University of Colorado. The Stanford Synchrotron Radiation Lightsource is supported by the US Department of Energy, Office of Science, Office of Basic Energy Sciences under contract no. DE-AC02-76SF00515. Jian Wang is thanked for providing support on STXM beamline SM at the Canadian Light Source (CLS). CLS is supported by the Canada Foundation for Innovation, Natural Sciences and Engineering Research Council of Canada, the University of Saskatchewan, the Government of Saskatchewan, Western Economic Diversification Canada, the National Research Council Canada, and the Canadian Institutes of Health Research. We are grateful to Professor Clara Chan (University of Delaware) for providing C K-edge STXM/XANES spectra of Fe-oxidizing bacteria extracellular structures, and to Prof John Spear (Colorado School of Mines), Dr Jennyfer Miot (Museum National D'Histoire Naturelle, Paris), Dr Jena Johnson (University of Colorado), Professor Yet-Ming Chiang (Massachusetts Institute of Technology) and three anonymous reviewers for their constructive comments on the manuscript.

## Author contributions
A.S.T. and J.C. contributed to the design of the study, interpretation of the results and preparation of the manuscript. J.C. performed the experiments and analysed the data.

## Additional information

**Competing financial interests:** The authors declare no competing financial interests.

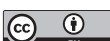

