## [Peer Review File · Nature Communications]

Reviewers' comments:

Reviewer #1 (Remarks to the Author):

I would like to see the authors address my comments which are centered around data presentation style and, more importantly, the applicability of their experiments to the natural environment.

A. Summary of the key results

The authors describe simple experiments in which they create opposite slope gradients of dissolved O₂ and sulfide in tubes. In the absence of organics, nothing happens. In the organic-rich case, structures grow that look like microbial structures. The authors make the argument that because of their morphology, distribution and composition, these structures could pass biogenicity tests. Such C/S structures forming in similarity abiotic conditions in nature could lead an observer to conclude that they are biogenic. I find the microscopic and chemical data presented here mostly validate the authors claim, although I have some methodological reservations.

This paper therefore aims to demonstrate that in certain conditions, structures that would be interpreted as biogenic are indeed not. This is an important result for the interpretation of the always putative fossil record of Earth's early microbial life.

The formation of S₀ abiotically, albeit in the presence of organics, challenges assumptions about the sulfur biogeochemical cycle. This is also another important observation, although the authors could address this point further, even if somewhat speculatively.

B. Originality and interest/G. References

I am very surprised by the omission of any reference to García-Ruiz et al., Science 2003, DOI: 10.1126/science.1090163. This must be corrected. The present draft is an obvious parallel to that earlier paper and they both come to similar conclusions in terms of interpretation of morphological and chemical possible microfossils. The target structures in each paper are different enough, so the Cosmidis and Templeton draft is definitely new results and deserves publication.

C. Data & methodology: validity of approach, quality of data, quality of presentation

- As a general comment for Figure 1. The photos are fine, but I feel the arrows and caption comments are not really enough to convey the authors' interpretation of what they show. An interpretative rendition of one (some) of the figure(s) would make it easier for the reader to see what the authors are trying to show. At least add a bit more labelling or commenting.
- Fig1f: I would rather see better resolved peaks for C, O, S than the flat spectra above 4KeV
- There is overlap in the purpose of figures 1 and 2, and the authors actually refer to both figures at the same time (L94-99). A better organization of the photos in Fig 1 and Fig2, maybe as one unique figure, would make the reading of the paper more linear.
- L113: In the absence of organics, S₀ occurs as rounded mineral grains... I would like to see a little bit more description about the morphological difference between this and the C/S spheres.
- Fig 3c: I find the spectroscopic data in suppl. Fig3 and 4 more specific and informative than in Fig3c. Consider including these data into the main part of the paper.
- Fig4e: hard to read, consider enlarging or color-coding

E. Conclusions: robustness, validity, reliability

- The authors argue around L174 that the experimental conditions are relevant to natural geochemical conditions. However, this point should be argued a little more. I would like to see quantitative comparison between the experimental conditions and natural conditions.

- One point in particular is important: the organic compound concentration. As the authors note around L92 there is a relationship between organic concentration and filament size. On L177, the organic concentration is significantly higher than in nature. This is fine as it is likely necessary to produce a wider range of C/S structures, but this should be discussed. With lower concentration of organics, smaller structures will form... how about in natural conditions?

- The authors rely for this on ref 27 Gleeson et al. This is an important point and I think it should be explained more clearly. Precisely what results from Gleeson et al. are supportive of the fact that similar C/S structures would form in natural concentration conditions? Furthermore, in the conclusion of their paper, Gleeson et al. write:

"Abiotic controls for our experiments did not show the same accumulations of sulfur; instead, limited abiotic precipitation only occurred in some high salinity and sulfide experiments, and the precipitates were highly crystalline and distinct in size and morphology from those of the inoculated experiments. Therefore, the distinct morphology of the biogenic sulfur structures suggests that biomineralization associated with sulfide-oxidizing bacteria has the potential to produce a morphological biosignature."

I think there is some contradiction here between the two papers that undermines the conclusion of the present draft. Please explain if and how this can be reconciled.

- A further point about the supplied organics: to what extent are the observed results dependent not only on the presence of mere organic compounds, which could be assumed to occur abiotically, but yeast extract, which implies ambient biological activity. This pertains to the argument that the C/S structures are false positives for biogenic structures, not about S₀ biogeochemistry. Could it be that they show abiotic S₀ formation, but still require some organic-rich milieu that would not be possible in abiotic conditions? How about organic sulfur in yeast extract? Is it possible to reproduce the results using abiotic organic compounds?

F. Suggested improvements: experiments, data for possible revision

Ideally, I would like to see the experiments replicated in natural, or plausible prebiotic concentrations of "abiotic organics". I admit this is a vague description, but at least experiments aiming in that direction could be designed. However, this is not in my view required for publication as long as the point in E are addressed properly.

H. Clarity and context

Besides comments above, this is OK.

Reviewer #2 (Remarks to the Author):

Cosmidis and Templeton show in this work that a variety of organic-encapsulated elemental sulfur nanostructures are formed at room temperature in gradient environments. These are fascinating results that point to a possible new means of generating sulfur nanostructures at room temperature for ion storage or other technological applications. Although I am not sufficiently familiar with the biogeochemical aspects to critically evaluate the novelty of this work in that context, I believe that this work will be of much interest to the energy research community, in which nanostructures of sulfur, silicon and carbon have been exploited for ion storage in recent years. This is a novel and apparently facile and economical approach to synthesizing such structures. I personally would be interested in testing the electrochemical storage properties of these materials.

Amongst the nanostructures shown, the helical structures are especially interesting. A mechanism for formation of the helical structures is not proposed in the manuscript. Do the authors have any proposed mechanisms, even speculative ones?

I believe this work is suitable for publication in Nature Communications, and that it would attract the attention of a broad readership.

Reviewer #3 (Remarks to the Author):

This manuscript describes the analysis of novel carbon and sulfur microstructures produced within sterile sulfide gradient tubes. Some of the filamentous and spherical microstructures resemble carbon and sulfur-containing structures preserved in the rock record that are sometimes interpreted as the remains of sulfur-metabolizing microbes. Overall, I found the manuscript to be novel, interesting, informative, and of excellent quality. In my opinion, the manuscript should be published in Nature Communications and would be of broad interest to your readership. However, the following comments should be addressed to improve the manuscript quality and rigor:

- General comment: One criticism of the establishment of these structures as pseudofossils comparable to those in the rock record concerns their size. It seems to me that the most "biological" of these structures are substantially smaller than those interpreted as fossilized microbes. The authors should clearly state how the size of these features compares with some of those structures interpreted as fossils in the rock record. If the features studied here are substantially smaller, then the authors should address this difference in the text.

. Line 49: The sentence: "The formation of these C/S microstructure furthermore provides a new mechanism for S₀ formation and stabilization involving complex interactions with organic matter that might shed light on S₀ formation processes in the environment, as well as a new pathway for the synthesis of nanostructured carbon/sulfur composites with the potential for industrial applications such as advanced cathode materials for Li-S batteries" is such a long sentence, and the synthetic nanostructure aspect of the paper really comes out of left field here. I would suggest that the authors split this long sentence into two sentences, and preface the second sentence with a sentence or two of context to set up the significance of the industrial applications of the C/S composites.

. Line 56 and results in general: Much of the methods/results is written in the present tense, while

convention would have these sections written in past tense (e.g., C/S microstructures were produced.)

. Line 58 and elsewhere: no need to capitalize the words "peptone" or "yeast extract".

. Line 59: should be "counter gradients of oxygen and sulfide were established"

. Line 233: It's an entirely reasonable argument that these carbon/sulfur microstructures should give us pause with respect to interpreting ancient sulfur-containing microstructures as fossils without additional evidence. But then here, the authors say that microstructures could be used as critical indicators of past environmental conditions. In the rock record, such features are either robust indicators of life (or environment) or they are ambiguous - I don't see how they can be both pseudofossils and specific indicators unless you can establish that life cannot produce structures like this. I would suggest removing this sentence, or adding something like "If criteria can be identified that allow us to rigorously differentiate abiotic from biotic C/S structures, then we might use them as critical indicators of such and such an environmental condition."

. Line 248: The authors state that "We show that nanostructured C/S composites can now be synthesized through a simple one-step energy-efficient reaction in sulfide gradient environments that mimic natural systems." I am not an expert in the area of synthetic nanotubes, but it doesn't seem to me that this manuscript presents evidence that these microstructures have characteristics that would make them suitable for industrial applications. Perhaps the authors could describe what characteristics are needed for C/S nanotubes to be useful in an industrial context and then explain how the microstructures they discovered meet these criteria (e.g., size, shape, chemistry etc.)

- The twisted stalk features in Figure 2g also resemble structures produced by Garcia-Ruiz, *Science* 323, 362 (2009). Although the composition is different, this previous discovery of a twisted stalk pseudofossil should be acknowledged by citing this earlier work and discussing it in the text.

Reviewer #4 (Remarks to the Author):

Elemental sulfur, often referred to as zerovalent sulfur, represents a complex suite of solids and dissolved compounds. Although known to humankind since biblical times, it's inorganic and biochemical chemistry continues to perplex and confound technical, microbiological and environmental scientists. Cosmidis and Templeton add a new layer of complexity to the sulfur story. This is a fascinating study, with implications for geochemistry, deep-time micropaleontology, as well as microbiology. The control of biomorphic mineral forms and their potential meaning for interpreting the rock-record is of course an important and timely cautionary tale for all of those working in deep-time/early earth microbiology. But just as important, the study highlights the surprising variability and potential for both biotic and abiotic control of mineral formation. I am reminded of the studies from David Rickard's group on the effect of aldehyde groups on the controlling the reaction of FeS and hydrogen sulfide to form pyrite, and the studies where he and his colleagues grew framboidal pyrite in celery husks (Rickard et al. *EPSL*, 2001; Sagemann et al., *Geochim. Cosmochim. Acta* 1999). Also related, and certainly worth mentioning in the manuscript, is the study by Prange et al., (*Microbiology*, 2002) where XANES is used to examine the zerovalent sulfur speciation in a number of sulfur bacteria. Prange et al. also observed sulfur globules with organic polysulfane type structures in *Allochromatium* species. It seems that microorganisms can direct the chemical architecture that Cosmidis and Templeton show can arise in an organic-rich matrix.

The authors use a full complement of state of the art methods. Specific comments and questions to some of the methods and interpretations are listed below. All in all, this is an elegant study that is sure to spur further research at the biotic-abiotic interface.

Specific comments:

The XANES/STXM data and presentation is a bit confusing. In the Methods description it is says that S K-edge data was collected (Line 335), but in Figure 3, S L-edge data is shown. On the other hand, in Figure 4, C K-edge data is shown, but is this from the SXTM as described in the Methods (Lines 346 and following), or actually from the XANES measurements made on the Stanford beamline (Line 336).

Lines 140-149: This is not exactly surprising, in that the authors are examining the basic composition of a yeast or other protein rich extract. If there are no organisms to degrade these compounds, I would expect many of them to remain behind, even if some react with polysulfides or sulfide. More interesting is whether or not the authors see any evidence for C-S bonds or disulfide bonds in an otherwise organic-C rich chemical environment. This would provide evidence of actual direct "cross-linking" as they speculate upon in line 160.

Lines 160 to 167: Supporting the idea that the organics are directly involved in the oxidation of the elemental sulfur is the observation that dissolved oxygen diffuses completely through to the bottom of the agar tube with evidence for consumption only at the beginning of the experiment. Furthermore, the "organic" profiles do not show the typical evidence of proton production that one finds with sulfide oxidation and is obvious in the blue "non-organic" curves. Of course one could argue that the organics strongly buffer the acid forming oxidation reactions.

Line 179: "Highly oligotrophic" does not mean low DOC; one can have relatively high DOC concentrations, yet be nutrient poor and unproductive. It would be interesting to see what happen in something like a fulvic acid rich environment.

Line 43: Please, very briefly define "dissolved organics", i.e. a mixture of organic compounds from dissolved yeast extract or peptone preparations.

Would it be possible to superimpose a typical H₂S/O₂ gradient on Figure 1a as measured by the microelectrodes (Methods, lines 273-286)?

Figure 4e: What do you mean Gallionella-like or Leptotrix-like? What does that "-like" mean?

Supplementary Figure 4a: What is the peak at ca. 350 cm⁻¹ for the no organic carbon control (b) experiment. This does not appear to be rhombic elemental S or polysulfide S. Please explain in the text or figure.

Manuscript NCOMMS-16-03827: “Self-assembly of biomorphic carbon/sulfur microstructures in sulfidic environments”

Responses to the Referees

We would like to thank the Referees for careful reading of the manuscript and their constructive comments. Changes and additions were made accordingly in the revised version of the manuscript. Responses to the Referees’ comments are provided below in italics. The line numbers refer to the revised version of the manuscript.

Reviewer #1

I would like to see the authors address my comments which are centered around data presentation style and, more importantly, the applicability of their experiments to the natural environment.

A. Summary of the key results

The authors describe simple experiments in which they create opposite slope gradients of dissolved O₂ and sulfide in tubes. In the absence of organics, nothing happens. In the organic-rich case, structures grow that look like microbial structures. The authors make the argument that because of their morphology, distribution and composition, these structures could pass biogenicity tests. Such C/S structures forming in similarity abiotic conditions in nature could lead an observer to conclude that they are biogenic. I find the microscopic and chemical data presented here mostly validate the authors claim, although I have some methodological reservations.

This paper therefore aims to demonstrate that in certain conditions, structures that would be interpreted as biogenic are indeed not. This is an important result for the interpretation of the always putative fossil record of Earth's early microbial life.

The formation of S⁰ abiotically, albeit in the presence of organics, challenges assumptions about the sulfur biogeochemical cycle. This is also another important observation, although the authors could address this point further, even if somewhat speculatively.

The manuscript does now contain more discussion about the importance of S⁰ in the sulfur biogeochemical cycle. For examples, please see more detailed responses below.

B. Originality and interest/G. References

I am very surprised by the omission of any reference to García-Ruiz et al., Science 2003, DOI: 10.1126/science.1090163. This must be corrected. The present draft is an obvious parallel to that earlier paper and they both come to similar conclusions in terms of interpretation of morphological and chemical possible microfossils. The target structures in each paper are different enough, so the Cosmidis and Templeton draft is definitely new results and deserves publication.

We agree that this paper is critical to highlight. We now cite and discuss García-Ruiz et al.’s work on silica-carbonate biomorphs (lines 255-263). Similar to their studies, the formation of our C/S “biomorphs” calls for greatest caution for the interpretation of morphological and chemical putative biosignatures in the rock record. However, we stress on the fact that García-Ruiz et al.’s

biomorphs were purely inorganic structures, whereas our C/S microstructures are organic-rich, making them even more relevant to the interpretation of putative microfossils.

C. Data & methodology: validity of approach, quality of data, quality of presentation

- As a general comment for Figure 1. The photos are fine, but I feel the arrows and caption comments are not really enough to convey the authors' interpretation of what they show. An interpretative rendition of one (some) of the figure(s) would make it easier for the reader to see what the authors are trying to show. At least add a bit more labelling or commenting.

The Figures have been significantly reorganized to address several of the Reviewers concerns and to better highlight the wealth of data and associated interpretations. To address this specific comment, we added some more discussion about Figure 1 in the main text in order to provide more interpretation of what is shown. We also wrote a more detailed caption for this figure, as well as for Figures 2 and 3.

- Fig1f: I would rather see better resolved peaks for C, O, S than the flat spectra above 4KeV

We understand this request and did explore using a different scale for the EDX spectra (e.g. only showing the data from 0-0.5eV). However, such a revised figure does not provide new information, yet it does lose information critical for the reader. It is important to show the flat spectra above 4KeV to demonstrate that there is nothing other than C, O and S that can be detected.

- There is overlap in the purpose of figures 1 and 2, and the authors actually refer to both figures at the same time (L94-99). A better organization of the photos in Fig 1 and Fig2, maybe as one unique figure, would make the reading of the paper more linear.

Figures 1 and 2 of the original manuscript contained a lot of information and several subparts. To minimize overlap and enhance the presentation of the data, we split out and rearranged these figures into five different figures (now Fig. 1-5), regrouping together images that were obtained using the same method. We also reorganized the text so that the figures are referred to and described in the correct order. These changes hopefully make the reading more linear, and more tightly correlated to the data, as well as making each image and caption more easily readable for the reader.

- L113: In the absence of organics, S⁰ occurs as rounded mineral grains... I would like to see a little bit more description about the morphological difference between this and the C/S spheres.

We have now added a SEM image of the S⁰ mineral grains formed in the absence of organics (Fig. 3a) and we now describe their morphology in the text (lines 93-95). The main differences between these grains and the spherical C/S microstructures is that they have irregular shapes (whereas most of the spherical C/S microstructures are perfectly spherical). Most important is the fact that they do not have an organic envelope around them, which is also emphasized in the text.

- Fig 3c: I find the spectroscopic data in suppl. Fig3 and 4 more specific and informative than in Fig3c. Consider including these data into the main part of the paper.

We are pleased to follow this suggestion and include these data in the main part of the paper. Figure 3c of the original manuscript (S L-edge STXM/XANES spectra) presented spectroscopic information about sulfur speciation acquired at a spatial scale of a few tens of nanometers. This figure showed that sulfur in the samples was present as zero-valent sulfur as also do Suppl. Figure 3 and Suppl. Figure 4 of the original manuscript (bulk S K-edge XANES and Raman spectromicroscopy). Since Suppl. Figure 3 of the original manuscript moreover provides a qualitative comparison of the amount of S^0 produced in the presence and absence of organics, this figure has been moved into the main part of the paper to illustrate the spectroscopic characterization of sulfur in the C/S microstructures instead of former Figure 3c. We also now further comment this new Figure (Figure 7) in the main text. Figure 3c of the original manuscript has been moved to Suppl. Figure 4.

- Fig4e: hard to read, consider enlarging or color-coding

We re-organized and enlarged the STXM images and spectra on this figure (now Fig. 8). We also added colors to the spectra and modified the energy scale (now 283-293 eV instead of 282-296 eV in the original manuscript) in order to make the important spectral features more visible.

E. Conclusions: robustness, validity, reliability

- The authors argue around L174 that the experimental conditions are relevant to natural geochemical conditions. However, this point should be argued a little more. I would like to see quantitative comparison between the experimental conditions and natural conditions.

We now more thoroughly demonstrate that dissolved sulfide concentrations in our experiments (a few hundreds of micromolar) can be found in environments where elemental sulfur forms, and provide examples (with references) of such environments (lines 174-178). We also reiterate that other critical parameters comparable to the environment are: simple mineral composition, circumneutral pH, oxic/anoxic interface, as explained in the discussion (lines 170-174).

- One point in particular is important: the organic compound concentration. As the authors note around L92 there is a relationship between organic concentration and filament size. On L177, the organic concentration is significantly higher than in nature. This is fine as it is likely necessary to produce a wider range of C/S structures, but this should be discussed. With lower concentration of organics, smaller structures will form... how about in natural conditions?

We now further discuss that the experimental conditions are indeed higher than typical environmental dissolved organic carbon concentrations (lines 178-179). At this stage, we can only speculate about what will happen under the lower natural conditions; such a discussion is more fully developed in the manuscript (lines 179-182).

- The authors rely for this on ref 27 Gleeson et al. This is an important point and I think it should be explained more clearly. Precisely what results from Gleeson et al. are supportive of the fact that similar C/S structures would form in natural concentration conditions? Furthermore, in the conclusion of their paper, Gleeson et al. write:

"Abiotic controls for our experiments did not show the same accumulations of sulfur; instead, limited abiotic precipitation only occurred in some high salinity and sulfide experiments, and the precipitates were highly crystalline and distinct in size and morphology from those of the inoculated experiments. Therefore, the distinct morphology of the biogenic sulfur structures suggests that biomineralization associated with sulfide-oxidizing bacteria has the potential to produce a morphological biosignature."

I think there is some contradiction here between the two papers that undermines the conclusion of the present draft. Please explain if and how this can be reconciled.

We have added content to the manuscript that more fully re-examines the findings from Gleeson et al., and reconciles their previous observations with this study (lines 200-218).

Gleeson et al. obtained filamentous microstructures mineralized with elemental sulfur in sulfide gradient tubes much similar to those used in our study, except that Gleeson et al. did not supplement them with dissolved organics such as yeast extract or peptone. Instead, they inoculated them with sediments from a natural environment (Borup Fiord Pass in the Arctic) or with microbial consortia enriched from the same environment. The filamentous structures they observe resemble some of the C/S microstructures we produce in our abiotic experiments. Gleeson et al. interpreted them as microbial biomineralized microbial filaments and sheaths, but we now recognize how they are identical to abiogenic C/S microstructures that can be formed without microbes. We thus suspect that C/S microstructures can be produced in the presence of very low organic concentrations, as the inoculated gradient tubes in Gleeson et al. only contained organics derived from Borup sediments, and/or organics excreted by the microorganisms (which could for example be derived from agar digestion by these organisms). However, as the organic carbon concentration in these experiments has not been measured, we no longer use this study to demonstrate C/S microstructures formation in the presence of low organic concentrations.

Gleeson et al. did not observe these microstructures in their non-inoculated tubes (the "abiotic controls" of their experiments). This can be explained by the fact that these abiotic tubes did not contain any source of dissolved organics.

Lastly, given that Gleeson et al. did interpret their filamentous microstructures as microbial, they proposed that they could be used as a morphological biosignature of sulfur-cycling bacteria. Thus we do use Gleeson et al.'s study to stress the fact that C/S microstructures formed in natural or environmental settings could be easily misinterpreted as microbial cellular or intracellular structures.

- A further point about the supplied organics: to what extent are the observed results dependent not only on the presence of mere organic compounds, which could be assumed to occur abiotically, but yeast extract, which implies ambient biological activity. This pertains to the argument that the C/S structures are false positives for biogenic structures, not about S⁰ biogeochemistry. Could it be that they show abiotic S⁰ formation, but still require some organic-rich milieu that would not be possible in abiotic conditions? How about organic sulfur in yeast extract? Is it possible to reproduce the results using abiotic organic compounds?

There are numerous open questions from our study about S^0 biogeochemistry, and we anticipate several follow up studies will ensue using a diversity of natural organic compounds of varying complexity, including +/- organic sulfur containing compounds. We are currently conducting experiments with sulfide gradient tubes amended with a range of simple organic compounds, including some that can be considered “prebiotic”, to address these types of questions. Such data will be included in future publications. For now, we do further discuss this point raised by the reviewer; we also elaborate upon the term “abiogenic” in the present study (lines 138-145).

F. Suggested improvements: experiments, data for possible revision

Ideally, I would like to see the experiments replicated in natural, or plausible prebiotic concentrations of "abiotic organics". I admit this is a vague description, but at least experiments aiming in that direction could be designed. However, this is not in my view required for publication as long as the point in E are addressed properly.

We agree with the reviewer that experiments conducted under lower organic carbon concentrations, and with “abiotic organics”, will be of great interest and we have initiated such work. However, we consider this very large set of ongoing experiments to be beyond the scope of this manuscript.

H. Clarity and context

Besides comments above, this is OK.

Reviewer #2

Cosmidis and Templeton show in this work that a variety of organic-encapsulated elemental sulfur nanostructures are formed at room temperature in gradient environments. These are fascinating results that point to a possible new means of generating sulfur nanostructures at room temperature for ion storage or other technological applications. Although I am not sufficiently familiar with the biogeochemical aspects to critically evaluate the novelty of this work in that context, I believe that this work will be of much interest to the energy research community, in which nanostructures of sulfur, silicon and carbon have been exploited for ion storage in recent years. This is a novel and apparently facile and economical approach to synthesizing such structures. I personally would be interested in testing the electrochemical storage properties of these materials.

Amongst the nanostructures shown, the helical structures are especially interesting. A mechanism for formation of the helical structures is not proposed in the manuscript. Do the authors have any proposed mechanisms, even speculative ones?

We appreciate the perspectives of the reviewer. To address questions about helical structures, we cite two review papers (References 46 and 47 of the revised manuscript) that describe the formation of organic tubes and spheres through the self-assembly of amphiphilic organic molecules. We propose in the discussion that this could be one of the possible mechanism for the self-assembly of the organic envelope of the C/S microstructures (in addition to cross-linking of the organic molecules by sulfur to form larger macromolecules). Reference 47 in particular mention that chiral amphiphilic molecules can self-assemble into helical tubes. The close-packing

of chiral molecules produces a bend that is at the origin of the helical morphology. We now describe this mechanism in the discussion (lines 283-285).

I believe this work is suitable for publication in Nature Communications, and that it would attract the attention of a broad readership.

We now discuss more extensively the potential use of the C/S microstructure as cathode materials for Li-S batteries (lines 287-326). We also modified the abstract to include this question (lines 18-19).

Reviewer #3

This manuscript describes the analysis of novel carbon and sulfur microstructures produced within sterile sulfide gradient tubes. Some of the filamentous and spherical microstructures resemble carbon and sulfur-containing structures preserved in the rock record that are sometimes interpreted as the remains of sulfur-metabolizing microbes. Overall, I found the manuscript to be novel, interesting, informative, and of excellent quality. In my opinion, the manuscript should be published in Nature Communications and would be of broad interest to your readership. However, the following comments should be addressed to improve the manuscript quality and rigor:

- General comment: One criticism of the establishment of these structures as pseudofossils comparable to those in the rock record concerns their size. It seems to me that the most "biological" of these structures are substantially smaller than those interpreted as fossilized microbes. The authors should clearly state how the size of these features compares with some of those structures interpreted as fossils in the rock record. If the features studied here are substantially smaller, then the authors should address this difference in the text.

It is correct that some of our C/S filamentous microstructures have diameters one or two orders of magnitude smaller than filamentous bacteria or microbial sheaths; however, several of the structures are microscale and easily misinterpreted as microbial filaments or sheaths. We now further explain that in our experiments, we observe a relationship between dissolved organics concentration and filaments thickness, so that these thinner microstructures are obtained in the presence of the lowest yeast extract and/or peptone concentrations, which also correspond to the conditions mimicking most closely natural environments (lines 246-254). However, such structures grow in size with time. In addition, we now explain that with other types of organics, this concentration-size relationship might be different.

. Line 49: The sentence: "The formation of these C/S microstructure furthermore provides a new mechanism for S^0 formation and stabilization involving complex interactions with organic matter that might shed light on S^0 formation processes in the environment, as well as a new pathway for the synthesis of nanostructured carbon/sulfur composites with the potential for industrial applications such as advanced cathode materials for Li-S batteries" is such a long sentence, and the synthetic nanostructure aspect of the paper really comes out of left field here. I would suggest that the authors split this long sentence into two sentences, and preface the second sentence with a sentence or two of context to set up the significance of the industrial applications of the C/S composites.

This sentence was split, and the technological potential of the C/S microstructures as composite cathode materials for Li-S batteries is now further explained in a separate paragraph (lines 53-61).

. Line 56 and results in general: Much of the methods/results is written in the present tense, while convention would have these sections written in past tense (e.g., C/S microstructures were produced.)

We now use the past tense to describe the experiments and their results in the main text and the Methods section.

. Line 58 and elsewhere: no need to capitalize the words "peptone" or "yeast extract".

“Yeast Extract” and “Peptone” were replaced by “yeast extract” and “peptone” throughout the text.

. Line 59: should be "counter gradients of oxygen and sulfide were established"

This was modified accordingly (line 67).

. Line 233: It's an entirely reasonable argument that these carbon/sulfur microstructures should give us pause with respect to interpreting ancient sulfur-containing microstructures as fossils without additional evidence. But then here, the authors say that microstructures could be used as critical indicators of past environmental conditions. In the rock record, such features are either robust indicators of life (or environment) or they are ambiguous - I don't see how they can be both pseudofossils and specific indicators unless you can establish that life cannot produce structures like this. I would suggest removing this sentence, or adding something like "If criteria can be identified that allow us to rigorously differentiate abiotic from biotic C/S structures, then we might use them as critical indicators of such and such an environmental condition."

We have clarified our intent here, to address the reviewers concerns and to avoid any confusion by future readers (lines 273-276). This study does stress that C/S microstructures should give us pause in interpreting ancient sulfur microstructures as fossils, and we do not seek to weaken this critical point. However, we do mention that information can be gleaned from the formation of a C/S structure – that organics and dissolved sulfide must be present, which is useful paleoenvironmental information. One must still establish alternative criteria to identify a microfossil appropriately.

. Line 248: The authors state that "We show that nanostructured C/S composites can now be synthesized through a simple one-step energy-efficient reaction in sulfide gradient environments that mimic natural systems." I am not an expert in the area of synthetic nanotubes, but it doesn't seem to me that this manuscript presents evidence that these microstructures have characteristics that would make them suitable for industrial applications. Perhaps the authors could describe what characteristics are needed for C/S nanotubes to be useful in an industrial context and then explain how the microstructures they discovered meet these criteria (e.g., size, shape, chemistry etc.)

Since there is sufficient room in the manuscript, we now further develop our perspectives on the application of C/S composites in energy storage technology (lines 287-326). The discussion was

enriched with an extended paragraph describing the desired properties of carbon-sulfur nanostructured composites as cathode materials for Li-S batteries, and describing how the C/S microstructures possess many characteristics that make them excellent candidate materials for this applications (in particular, confinement of sulfur in carbon tubes and sphere with high surface area, potential for accommodating volume expansion, high sulfur loading, and tunability), while being produced through a much more effective low-temperature, low-cost method compared with existing synthesis methods.

- The twisted stalk features in Figure 2g also resemble structures produced by Garcia-Ruiz, Science 323, 362 (2009). Although the composition is different, this previous discovery of a twisted stalk pseudofossil should be acknowledged by citing this earlier work and discussing it in the text.

We now cite and discuss García-Ruiz et al.'s silica-carbonate biomorphs in the discussion (lines 255-258). In particular, we stress on the resemblance of their helical biomorphs with our twisted C/S microstructures.

Reviewer #4

Elemental sulfur, often referred to as zerovalent sulfur, represents a complex suite of solids and dissolved compounds. Although known to humankind since biblical times, its inorganic and biochemical chemistry continues to perplex and confound technical, microbiological and environmental scientists. Cosmidis and Templeton add a new layer of complexity to the sulfur story. This is a fascinating study, with implications for geochemistry, deep-time micropaleontology, as well as microbiology. The control of biomorphic mineral forms and their potential meaning for interpreting the rock-record is of course an important and timely cautionary tale for all of those working in deep-time/early earth microbiology. But just as important, the study highlights the surprising variability and potential for both biotic and abiotic control of mineral formation. I am reminded of the studies from David Rickard's group on the effect of aldehyde groups on the controlling the reaction of FeS and hydrogen sulfide to form pyrite, and the studies where he and his colleagues grew framboidal pyrite in celery husks (Rickard et al. EPSL, 2001; Sagemann et al., Geochim. Cosmochim. Acta 1999). Also related, and certainly worth mentioning in the manuscript, is the study by Prange et al., (Microbiology, 2002) where XANES is used to examine the zerovalent sulfur speciation in a number of sulfur bacteria. Prange et al. also observed sulfur globules with organic polysulfane type structures in *Allochromatium* species. It seems that microorganisms can direct the chemical architecture that Cosmidis and Templeton show can arise in an organic-rich matrix.

The authors use a full complement of state of the art methods. Specific comments and questions to some of the methods and interpretations are listed below. All in all, this is an elegant study that is sure to spur further research at the biotic-abiotic interface.

We now cite Rickard et al., EPSL, 2001 (Reference 28) in the discussion, as another example of a study showing an influence of organics on the sulfide oxidation reaction (lines 159-161).

We also cite Prange et al., Microbiology, 2002 (Reference 33) in order to stress on the fact that microorganisms might control the formation of intracellular sulfur through interactions with organics (lines 190-191).

Specific comments:

The XANES/STXM data and presentation is a bit confusing. In the Methods description it is says that S K-edge data was collected (Line 335), but in Figure 3, S L-edge data is shown. On the other hand, in Figure 4, C K-edge data is shown, but is this from the SXTM as described in the Methods (Lines 346 and following), or actually from the XANES measurements made on the Stanford beamline (Line 336).

We now make more clearly the distinction between bulk S K-edge XANES spectra which were obtained at Stanford Synchrotron Radiation Lightsource and STXM-based XANES spectra (at the C K-edge and S L-edge) which were obtained at the Canadian Light Source, by calling the former XANES spectra and the latter STXM/XANES spectra.

Lines 140-149: This is not exactly surprising, in that the authors are examining the basic composition of a yeast or other protein rich extract. If there are no organisms to degrade these compounds, I would expect many of them to remain behind, even if some react with polysulfides or sulfide. More interesting is whether or not the authors see any evidence for C-S bonds or disulfide bonds in an otherwise organic-C rich chemical environment. This would provide evidence of actual direct "cross-linking" as they speculate upon in line 160.

We now better emphasize our assessment of this point in the discussion (lines 164-167): ‘The presence of organosulfur compounds was not detected in our experiments by bulk S K-edge XANES or S L-edge STXM/XANES analyses, either because they are not present in appreciable quantities, or because the total sulfur pool is dominated by the S^0 co-precipitated in the C/S microstructures.’ We can thus only speculate that cross-linking of organics by sulfur might occur.

Lines 160 to 167: Supporting the idea that the organics are directly involved in the oxidation of the elemental sulfur is the observation that dissolved oxygen diffuses completely through to the bottom of the agar tube with evidence for consumption only at the beginning of the experiment. Furthermore, the "organic" profiles do not show the typical evidence or proton production that one finds with sulfide oxidation and is obvious in the blue "non-organic" curves. Of course one could argue that the organics strongly buffer the acid forming oxidation reactions.

We now further discuss the oxygen profiles in the gradient tubes in the caption of Suppl. Figure 1. Oxygen diffusion in the “organic” tubes after 6 days of experiment is indeed an indication that sulfide has been consumed. We don’t discuss the pH profiles here as the buffering effect of the organics cannot be assessed. In addition, proton production is not expected to be nearly so significant when S^0 is the product.

Line 179: "Highly oligotrophic" does not mean low DOC; one can have relatively high DOC concentrations, yet be nutrient poor and unproductive. It would be interesting to see what happen in something like a fulvic acid rich environment.

We modified this part of the discussion by further describing previous organic-poor sulfide gradient tubes experiments by Gleeson et al., Geobiology, 2011 (lines 200-218), and we no longer use the term “oligotrophic”.

A future article describing new experiments currently being conducted will describe C/S microstructures formation in the presence of diverse simple organic compounds as well as natural organic matter such as humic acids.

Line 43: Please, very briefly define "dissolved organics", i.e. a mixture of organic compounds from dissolved yeast extract or peptone preparations.

“Dissolved organics” is now briefly described in the introduction, as “complex mixtures of organic compounds such as yeast extract and/or peptone” (lines 42-43).

Would it be possible to superimpose a typical H₂S/O₂ gradient on Figure 1a as measured by the microelectrodes (Methods, lines 273-286)?

We modified Fig. 1 by adding typical H₂S and O₂ profiles in the top layer of the tubes after one day.

Figure 4e: What do you mean Gallionella-like or Leptothrix-like? What does that "-like" mean?

The STXM/XANES C K-edge spectra corresponding to extracellular structures from “Gallionella-like” and “Leptothrix-like” bacteria were provided by Clara S. Chan, and have previously been published in Chan et al., Geochimica et Cosmochimica Acta 73 (2009) (Ref. 21 of the original manuscript). These spectra were obtained on samples collected from a natural environment (Piquette Mine), where extracellular microbial structures resembling those produced by Gallionella and Leptothrix were found. However, in this study, no complementary analyses (for instance, genomic) have been performed that would have allowed assigning unambiguously those structures to Gallionella and Leptothrix. Clara Chan and co-authors thus describe them as “sheaths reminiscent of the Fe-oxidizing bacterium Leptothrix ochracea and twisted stalks reminiscent of the Fe-oxidizing bacterium Gallionella ferruginea”. We now specify in the Methods section:

“The C K-edge spectra of the samples were compared with spectra obtained on E. coli cells, Mariprofundus ferrooxydans stalks, as well as environmental twisted stalks reminiscent of Gallionella ferruginea (“Gallionella-type stalks”) and sheaths reminiscent of Leptothrix ochracea (“Leptothrix-type sheaths”).” (lines 450-454)

Supplementary Figure 4a: What is the peak at ca. 350 cm⁻¹ for the no organic carbon control (b) experiment? This does not appear to be rhombic elemental S or polysulfide S. Please explain in the text or figure.

The peak at ~322 cm⁻¹ is the main peak in the Raman spectrum of the CaF₂ plates on which the samples were deposited, as explained in the Methods section. This peak is present in the spectra of all samples, but is much taller (compared with the elemental sulfur peaks) in the No organic carbon control sample compared with the samples prepared with organics, as this control contains less S⁰. We added in the caption of Supplementary Fig. 4a (now Supplementary Fig. 3a): “The

peak at $\sim 322\text{ cm}^{-1}$ corresponds to the CaF_2 substrate on which the samples were deposited for Raman analyses.”

Reviewers' comments:

Reviewer #1 (Remarks to the Author):

In this new revised version of their manuscript, the authors have substantially reorganized their figures. They have refined their arguments about the "abiotic" nature of their experiments. They have also acknowledged the need for more experiments. These were my main concerns about the original manuscript and I'm happy to see the changes made. I believe the paper is now easier to read, more concise and more aware of its limitations. I gladly recommend its publication.

Two small remarks:

- because the narrative is clearer, they lack/need for "really abiotic" condition experiments is even more obvious. It's ok if these are not performed for this manuscript, and the authors have adjusted the scope of their conclusions accordingly.
- about the "energy research" impact, following the remarks of reviewer #2 (I know nothing about this field). It's only mentioned in the last line of the abstract, but it takes a significant part of the main text. This feels a little imbalanced. It is interesting, and difficult, to mix biogeochemistry and energy science in a single paper, but it hurts the coherence of the paper somewhat. I'll defer to the other reviewers about this.

Reviewer #2 (Remarks to the Author):

I have reviewed the rebuttal by the authors, and certainly their responses to my (relatively minor) comments were adequate. The other reviewers posed considerably more detailed questions to be answered, but I thought that the authors' responses to those comments were also sufficient to warrant acceptance of the manuscript. So I am in favor of acceptance of the paper.

Reviewer #3 (Remarks to the Author):

I have now reviewed the revised manuscript and conclude that the authors have adequately addressed all of the concerns and questions I had about the previous version of the manuscript. As such, I now strongly recommend the manuscript for publication in Nature Communications.

Reviewer #4 (Remarks to the Author):

The authors have done an excellent job of responding to my questions and comments. I appreciate the effort they took to incorporate suggestions into the manuscript. They have improved and polished what was already a fascinating report. I look forward seeing its publication in Nature Communications.

One tiny quibble: In figures 1 and S1, total dissolved sulfide concentrations are referred to in the legend as " S^{2-} ", when they actually measured H_2S and calculated the sum of H_2S and HS^- and these are the compounds involved in any reaction. (S^{2-} exists in trivial concentrations.) Could the authors somehow mention this in the captions?

Manuscript NCOMMS-16-03827: “Self-assembly of biomorphic carbon/sulfur microstructures in sulfidic environments”

Responses to the Referees

We would like to thank the Referees for their positive reception of the revised manuscript. There was only one suggested edit, by Reviewer #4, and we have accordingly clarified the calculation of S^{2-} presented in Figure 1 and S1.

Reviewer #1:

In this new revised version of their manuscript, the authors have substantially reorganized their figures. They have refined their arguments about the "abiotic" nature of their experiments. They have also acknowledged the need for more experiments. These were my main concerns about the original manuscript and I'm happy to see the changes made. I believe the paper is now easier to read, more concise and more aware of its limitations. I gladly recommend its publication.

Two small remarks:

- because the narrative is clearer, they lack/need for "really abiotic" condition experiments is even more obvious. It's ok if these are not performed for this manuscript, and the authors have adjusted the scope of their conclusions accordingly.

- about the "energy research" impact, following the remarks of reviewer #2 (I know nothing about this field). It's only mentioned in the last line of the abstract, but it takes a significant part of the main text. This feels a little imbalanced. It is interesting, and difficult, to mix biogeochemistry and energy science in a single paper, but it hurts the coherence of the paper somewhat. I'll defer to the other reviewers about this.

Reviewer #2:

I have reviewed the rebuttal by the authors, and certainly their responses to my (relatively minor) comments were adequate. The other reviewers posed considerably more detailed questions to be answered, but I thought that the authors' responses to those comments were also sufficient to warrant acceptance of the manuscript. So I am in favor of acceptance of the paper.

Reviewer #3:

I have now reviewed the revised manuscript and conclude that the authors have adequately addressed all of the concerns and questions I had about the previous version of the manuscript. As such, I now strongly recommend the manuscript for publication in Nature Communications.

Reviewer #4:

The authors have done an excellent job of responding to my questions and comments. I appreciate the effort they took to incorporate suggestions into the manuscript. They have improved and polished what was already a fascinating report. I look forward seeing its publication in Nature Communications.

One tiny quibble: In figures 1 and S1, total dissolved sulfide concentrations are referred to in the legend as " S^{2-} ", when they actually measured H_2S and calculated the sum of H_2S and HS^- and these are the compounds involved in any reaction. (S^{2-} exists in trivial concentrations.) Could the authors somehow mention this in the captions?

The following statement was added to the caption of Figure 1 and Supplementary Figure 1: "Total sulfide concentrations were calculated using measured H_2S and pH profiles (see Methods section)."